# Side-Chain-Assisted Transition of Conjugated Polymers from a Semiconductor to Conductor and Comparison of Their NO_2_ Sensing Characteristics

**DOI:** 10.3390/ma16072877

**Published:** 2023-04-04

**Authors:** Yejin Ahn, Sooji Hwang, Hyojin Kye, Min Seon Kim, Wi Hyoung Lee, Bong-Gi Kim

**Affiliations:** Department of Organic and Nano System Engineering, Konkuk University, 120 Neungdong-ro, Seoul 05029, Republic of Korea

**Keywords:** conjugated polymer, electrical properties, side chain effect, resistive sensor, field effective sensor, NO_2_ detection

## Abstract

To investigate the effect of a side chain on the electrical properties of a conjugated polymer (CP), we designed two different CPs containing alkyl and ethylene glycol (EG) derivatives as side chains on the same conjugated backbone with an electron donor-acceptor (D-A) type chain configuration. **PTQ-T** with an alkyl side chain showed typical *p*-type semiconducting properties, whereas **PTQ-TEG** with an EG-based side chain exhibited electrically conductive behavior. Both CPs generated radical species owing to their strong D-A type conjugated structure; however, the spin density was much greater in **PTQ-TEG**. X-ray photoelectron spectroscopy analysis revealed that the O atoms of the EG-based side chains in **PTQ-TEG** were intercalated with the conjugated backbone and increased the carrier density. Upon application to a field-effect transistor sensor for **PTQ-T** and resistive sensor for **PTQ-TEG**, **PTQ-TEG** exhibited a better NO_2_ detection capability with faster signal recovery characteristics than **PTQ-T**. Compared with the relatively rigid alkyl side chains of **PTQ-T**, the flexible EG-based side chains in **PTQ-TEG** have a higher potential to enlarge the free volume as well as improve NO_2_-affinity, which promotes the diffusion of NO_2_ in and out of the **PTQ-TEG** film, and ultimately resulting in better NO_2_ detection capabilities.

## 1. Introduction

Conjugated polymers (CPs) are promising active materials that can be used in various electronic devices, such as organic photovoltaic cells, field-effect transistors (FETs), and thermoelectric devices [1,2,3,4,5,6]. The electrical properties of CPs strongly depend on their chemical structures, which determine the degree of *p*-orbital overlap in the conjugated frameworks [7,8,9]. Recently, the insertion of electron-accepting dopants into electron-donating semiconductors has been reported to efficiently induce electron transfer or form a charge transfer (CT) complex, resulting in electrical conductors [10]. However, because infinite *p*-orbital overlap through conjugated structures is entropically unfavorable, most linear CPs exhibit a kinked chain configuration, resulting in semiconducting properties unless charge carriers are intentionally generated via chemical or electrical doping processes [11,12,13,14]. Few examples of CPs exhibiting essentially conductor-like properties have been reported. As a result, the design strategies for devising conductive CPs have not been fully established.

CPs have received great attention as versatile gas detection platforms because of their ease of structural modification to improve the detection suitability for target gases [15,16,17]. For example, NO_2_ is a toxic gas that is abundantly released from industrial sources, and its strong oxidizing properties can critically damage the human respiratory system. Therefore, the accurate and fast detection of NO_2_ is necessary to ensure workplace safety. Recently, our research group reported that EG-based side chains have good affinity for polar NO_2_ molecules, providing better and faster NO_2_ detection properties [18]. The electronic performance of CP can be dramatically altered by side-chain engineering without modifying the conjugated backbone [19,20,21,22,23]. Therefore, the design of a side chain capable of specific interaction with an analyte to improve the detection sensitivity of CP-based electronic sensors is in high demand. In addition, since the porous morphology of CP can facilitate gas diffusion and enlarge the reactive surface area [24,25,26,27,28], CP films with porous structures have been reported to provide enhanced gas detection capabilities [29,30].

CP-based electronic sensors have two different device configurations: chemiresistive- and FET-type sensors. Chemiresistive sensors utilize a change in resistance to detect the target gas; therefore, CP as a detection platform should have electrical conductor-like properties [31,32,33]. FET-based sensors, on the other hand, adopt semiconducting CPs to monitor current changes due to charge carrier mobility and/or concentration, which enable the delivery of multiparameter response characteristics, such as charge–carrier mobility, threshold voltage, on/off current, and conductivity [34,35,36]. Various CPs have been implemented in chemiresistive- and FET-type electronic sensors, and their advantages have been reported in terms of detection sensitivity, selectivity, and fast recovery. For example, an ion-in-conjugation polymer, *p*-polyphenyl squaraine (*p*-PPS), has been adopted as a chemiresistive sensor for NO_2_ gas detection and exhibited a high sensitivity of 100 ppb with a detection limit of 40 ppt [37]. In another example, a diketopyrrolopyrrole (DPP)-based CP bearing NO_2_-affinitive ethylene glycol (EG) side chains has registered sub-ppm detectivity when adopted in FET-type sensors [18]. However, it is unreasonable to directly compare the gas detection performances of chemiresistive- and FET-type sensors because their working mechanisms and structures of the applied polymers are different.

In this study, we designed two different CPs that share the same thiadiazolo–quinoxaline-based conjugated backbone with different side chains. In the designed CPs, alkyl- and EG-based side chains were introduced into the same quinoxaline moiety, and their electrical properties were characterized using a FET-based electrode configuration. While **PTQ-T** containing alkyl side chains exhibited typical *p*-type semiconductor-like behavior, **PTQ-TEG** having EG-based side chains showed a conductor-like current-voltage (*I*-*V*) response. The difference in the electrical properties of the two CPs was discussed by comparing their molecular properties, such as molecular energy level, film morphology, and atomic binding energy. Finally, the NO_2_ detection performances of **PTQ-T** and **PTQ-TEG** were compared by applying them to chemiresistive- and FET-type sensors, respectively. Although both CPs have similar chemical structure, it was found that **PTQ-TEG** implemented in the chemiresistive sensor marked higher NO_2_ sensitivity and fast recovery characteristics.

## 2. Materials and Methods

### 2.1. Materials

The reagents used in this study were purchased from commercial suppliers (Sigma-Aldrich, Tokyo Chemical Industry, and Alfa Aesar) and used without further purification. Compounds **1a** and **1b** were synthesized similarly to the previously reported methods [18,38]. The chemical structure of CP was confirmed using Fourier transform infrared (FT-IR) and proton nuclear magnetic resonance (^1^H-NMR) spectroscopy (Appendix A). The molecular weight and polydispersity index (PDI) were determined by gel permeation chromatography with chloroform as an eluent (Appendix A).

### 2.2. PTQ-T and PTQ-TEG

In a dried Schlenk flask, **1a** (0.5 g, 0.40 mmol) and 2,5-bis(trimethylstannyl)thiophene (0.16 g, 0.40 mmol) were dissolved in anhydrous xylene (2 mL) under an argon atmosphere. After adding tris(dibenzylideneacetone)dipalladium (2.2 mol%) and tris(*o*-tolyl) phosphine (3.2 mol%), the mixture was stirred in a microwave reactor at 140 ℃ for 2 h. Then, 2-tributylstannyl thiophene (5 mol%) and 2-bromothiophene (5 mol%) were added at 30 min intervals to terminate polymerization. The solution was then poured into methanol, and the precipitate was collected using a nylon membrane filter. To remove the low molecular weight portion, the obtained solid was further purified by Soxhlet extraction using methanol, acetone, hexane, and chloroform. The portion soluble only in chloroform was reprecipitated in methanol, and the final **PTQ-T** was collected in 75% yield by filtration using a nylon membrane: Mw (18.7 kDa) and PDI (3.10). Similar to the method for the polymerization of **PTQ-T**, **PTQ-TEG** was obtained in 68% yield through copolymerization with **1b** (0.5 g, 0.40 mmol) and 2,5-bis(trimethylstannyl)thiophene (0.16 g, 0.398 mmol): Mw (16.3 kDa) and PDI (2.52).

### 2.3. FET and Gas Sensor Fabrication

A heavily n-doped silicon wafer containing a SiO_2_ layer (300-nm-thick with a capacitance of 10.8 nF/cm^2^, Fine Science) was cut into 2 cm × 2 cm pieces. After cleaning the pieces by ultrasonication in acetone and isopropyl alcohol for 20 min, they were dried under N_2_ flow and then subjected to a 20 min UV–ozone treatment to modify the silicon wafer surface. After forming a self-assembled monolayer of octadecyl trichlorosilane on the substrate, each CP solution dissolved in chloroform (5 mg/mL) was spin-cast onto the self-assembled monolayer at 1500 rpm for 60 s. The formed CP film was dried in a vacuum oven at 25 °C for 4 h to remove residual solvent. To investigate the electrical properties of obtained CP thin films (57.0 ± 5.0 nm), source and drain electrodes (Au, 50 nm) with a channel length of 100 µm and a width of 2000 µm were thermally deposited on the CP layer. The FET active channel was placed in a gas sensor device and wired with Ag wire.

### 2.4. Electrical and Gas Sensing Properties

The electrical characteristics of the FET devices were measured using a Keithley 4200-SCS semiconductor parameter analyzer (Keithley, Cleveland, OH, USA) connected to a probe station (MS TECH, Suwon, Republic of Korea). To obtain the transfer curve, the gate bias (V_G_) was swept from 40 V to −80 V in −1.0 V increments, while the source-drain voltage (V_DS_) was fixed at −80 V. The output curve was collected by sweeping V_DS_ from 0 to −80 V in −1.0 V increments while fixing V_G_ at 0 V, −20 V, −40 V, −60 V, and −80 V, respectively. The gas-sensing properties of CPs were measured using a gas sensor (Precision Sensor System Inc., Daejeon, Republic of Korea). The applied voltage (V_G_ and V_DS_) was fixed at −10 V in the FET-type sensor, and the resistor-type sensor applied V_DS_ (−10 V) at zero V_G_ to measure the sensitivity to the analyte gas. The gas detection sensitivity was identified as the average value through repeated exposure to the analyte gas (50 ppm) for 50 s and filling with N_2_ three times for 900 s.

### 2.5. Characterization

Polystyrene was used as a standard for gel permeation chromatography (Waters, Worcester County, MA, USA) to determine the molecular weights of the obtained CPs. UV-visible (UV-vis) absorption spectra of CP were characterized using UV-vis spectroscopy (Agilent, Santa Clara, CA, USA) in solution (chloroform) and film states, respectively. Ultraviolet photoelectron spectroscopy (UPS, Riken, Tokyo, Japan) was performed to determine the energy levels of the CPs. The surface morphologies of CP thin films were characterized using atomic force microscopy (AFM, Park Systems, Suwon, Republic of Korea) in a non-contact mode. The degree of CP chain assembly in the film state was analyzed using 2D-GIXRD (Xenocs, Grenoble, France), and the atomic binding energies of the CPs were determined using X-ray photoelectron spectroscopy (XPS, Thermo-Fisher, Seoul, Republic of Korea). All measurements using the CP thin films were performed with the same thickness (57.0 ± 5.0 nm) applied to the gas sensor device.

## 3. Results and Discussion

As shown in Appendix A and Figure 1a, two CPs containing the same thiadiazolo-quinoxaline-based conjugated backbone but different side chains were designed to compare their electrical properties. Both CPs were obtained through a Stille-type cross-coupling reaction between thiadiazolo-quinoxaline and thiophene monomers, and their chemical structures were confirmed using FT-IR and ^1^H-NMR spectroscopy. Both CPs exhibited the same skeletal vibrations corresponding to thiophene (1603 cm^−1^) and quinoxaline (1513 cm^−1^) moieties in the FT-IR spectra. However, the stretching vibrations corresponding to C-H (2700–3000 cm^−1^) and C-O-C (1025–1100 cm^−1^) were clearly distinguished (Appendix A) because the side chains introduced into each CP are different. In addition, judging from the peak position and the integral ratio of aromatic and aliphatic protons in the ^1^H-NMR spectra (Appendix A), it was confirmed that polymerization was carried out successfully. Specifically, the integral ratio of aromatic and aliphatic protons was 1:7.1 for **PTQ-T** and 1:4.5 for **PTQ-TEG**, which was consistent with the theoretical values at a similar level (1:5.9 for **PTQ-T** and 1:4.1 for **PTQ-TEG**). The electrical properties of CPs have been known to be partially affected by the side chain. For example, the aggregation and chain assembly propensities of CPs depend on the side chains introduced onto the conjugated backbone, which critically affect the charge-carrier mobility of the CPs [7]. To compare the electrical properties of the CPs, alkyl and flexible EG-based side chains were introduced into the same conjugated backbone. In the conjugated framework of the obtained CPs, the thiadiazolo-quinoxaline derivative is a strong electron-accepting moiety, and the connected thiophenes have electron-donating characteristics. Therefore, the obtained CPs have an electron donor-acceptor (D-A) chain configuration. The typical characteristics of D-A type CPs include a bimodal-shaped absorption spectrum and narrow energy band-gap [39,40]. It has been known that, in the bimodal-shaped absorption spectrum of the D-A type CPs, the absorption in the shorter wavelength region is from π–π* transitions, and the red-shifted absorption originates from the intramolecular charge transfer (CT) between electron donating and accepting moieties that are covalently interconnected in the conjugated skeleton [41].

As shown in Figure 1b, when the absorption tendencies of the obtained CPs were characterized using UV-vis absorption spectroscopy, although both CPs have the same conjugated backbone, **PTQ-T** exhibited more red-shifted absorption than **PTQ-TEG** in both solution and film states. In general, alkyl side chains have a relatively rigid nature compared to EG-based side chains due to higher rotation barrier energy. Therefore, introducing alkyl side chains into CP (**PTQ-T** in this study) is likely to enhance the chain stiffness, which can promote non-covalent interactions between the CP chains and result in red-shifted absorption compared to analogous CPs with EG-based side chains [34,42]. Interestingly, although both CPs exhibited bimodal-shaped absorption spectra commonly seen in D-A type CPs [43,44], the absorption in the film state was found to be more red-shifted than that in the solution state, only in the longer wavelength region. This can be attributed to facile intramolecular CT interactions between D-A moieties via partial chain planarization or enhanced non-covalent interactions of CP chains in the film state [45]. When the highest occupied molecular orbital (HOMO) and lowest unoccupied molecular orbital (LUMO) of each CP were identified using AC2 and the absorption edge, both CPs were confirmed to have similar molecular energy levels with an extremely narrow band-gap of 0.9 eV (Figure 1c). The narrow band-gap in both CPs implies the strong electron-accepting property of the thiadiazolo-quinoxaline unit because the band-gap of quinoxaline-based CPs with similar structures commonly exceeds 1.0 eV [46,47].

Although **PTQ-T** and **PTQ-TEG** share the same conjugated framework, they exhibited completely different electrical properties. To compare the electrical properties of the CPs, a FET device with a bottom-gate top-contact structure was fabricated. The **PTQ-T**-based FET showed a typical *p*-type charge-transport behavior, and the hole mobility was determined to be 0.002 cm^2^/V·s from the charge-carrier transfer curve (Figure 2a and Appendix A). However, as shown in Figure 2b, **PTQ-TEG** exhibited a considerably high current level with no off-current region in the V_G_ sweep from −80 V to 40 V. In the output curve, even when V_G_ was 0 V, **PTQ-TEG** exhibited a proportional trend in the output current with increasing V_DS_ (Appendix A), indicating electrically conducting characteristics. Typically, the electrical conductivity of semiconducting CPs does not exceed 10^−7^ S/cm because of the insufficient charge-carrier density unless they are chemically or electrically doped [48]. However, the obtained conductivity of **PTQ-TEG** was 3.0 × 10^−4^ S/cm, which is too high of a value for **PTQ-TEG** to be labeled as a conventional semiconducting CP.

The distinct electrical properties of **PTQ-TEG**, compared to **PTQ-T,** could be due to the contribution of EG-based side chains because both CPs have the same conjugated framework. To ascertain the effect of the EG-based side chain on the electrical properties of **PTQ-TEG**, the film morphologies and chain assembly features of both CPs were characterized using AFM and 2D-GIXRD, respectively, because the electrical properties of CPs are known to be sensitively affected by the degree of chain aggregation and assembly [49,50]. As shown in Figure 3a,b, both CPs exhibited featureless smooth surfaces with root-mean-square (RMS) roughness values of 0.478 nm and 0.355 nm for **PTQ-T** and **PTQ-TEG**, respectively. Compared to **PTQ-T**, **PTQ-TEG** showed a slightly reduced RMS roughness. The reduced RMS roughness can be attributed to the enhanced fluidity of the EG-based side chain, which has a lower rotational barrier energy than the alkyl side chain. In addition, the chain assembly tendencies of the CPs in the film state were examined using 2D-GIXRD. As shown in Figure 3c,d, both CPs exhibit diffused diffraction tendencies, indicating amorphous-like weak crystalline characteristics. When comparing the diffraction intensities in the low-*q* region of both CPs (Appendix A), the diffraction pattern in the in-plane direction (*xy*-axis, *q*_xy_) was confirmed to be more prominent than that in the out-of-plane direction (*z*-axis, *q*_z_), implying that both CPs prefer face-on-type chain assemblies in the film state. AFM and 2D-GIXRD measurements indicated that the CPs had similar morphologies, including a chain assembly tendency, which implies that the difference in the electrical properties between **PTQ-T** and **PTQ-TEG** did not originate from the film morphology.

To confirm whether the difference in the electrical properties of **PTQ-T** and **PTQ-TEG** was related to their inherent chemical structures, the binding energies of the elements present in the CPs were analyzed using XPS. As shown in Figure 4, when the binding energies of the C, N, and O atoms were compared, the binding-energy patterns of the N atoms were almost the same in both CPs; however, the binding energies of C and O were clearly distinguished. As shown in Figure 4a, the binding-energy difference between the C atoms can be inferred to be due to the C atoms in the EG-based side chain introduced in **PTQ-TEG**. The C atom bonded to the O atom in the EG-based side chain of **PTQ-TEG** partially loses electrons and shows a relatively high binding energy because the O atom is more electronegative than the C atom. However, it can be clearly noted that the binding energy of O atoms in **PTQ-TEG** has shifted to the higher binding-energy region compared to **PTQ-T**. Upon further analysis of the binding-energy distribution of the O atoms in **PTQ-TEG**, the binding-energy spectrum could be separated into two different types of O atoms. One type showed the same binding energy as **PTQ-T**, whereas the other type had a higher binding energy. Furthermore, as depicted in Figure 4b, the O atoms with a higher binding energy (C-O*) occupied a larger portion than the neutral O atoms (C-O) in **PTQ-TEG**. The shift to a higher binding energy for the O atoms implies that they donate electrons. Although the exact interpretation of the binding-energy shift is limited, charge carriers were likely to be generated, because of electron donation by the O atoms in the EG-based side chains, resulting in the electrical conducting properties of **PTQ-TEG**.

To compare the carrier densities quantitatively, the polaron spin densities of both CPs were measured using electron spin resonance (ESR). As shown in Figure 5, both CPs clearly show similar ESR signals without any artificial charge generation treatment, such as doping. Radical cations and anions have been reported to be generated by CT interaction when strong electron donating and accepting molecules are electronically intercalated [10]. Radical generation via CT interactions can also occur when D-A type structures form within a conjugated framework. For example, when pure organic materials contain strong electron acceptors in their conjugated structures, radicals have been generated via CT interactions and exhibit organic magnetism [51,52]. Therefore, it can be speculated that the ESR signals in both CPs originate from the strong intramolecular CT interaction between the strong electron accepting thiadiazolo-quinoxaline and electron donating thiophene, because similar ESR signals appear in both CPs with the same conjugated framework. In addition, the asymmetric ESR signals appearing for both CPs indicate that they are the result of multiple radical species rather than a single type of radical. When further analyzed using a Lorentz fitting, the ESR signal was deconvoluted into two different ESR signals with g-factors of 2.0100 and 2.0027. A g-factor of 2.0100 is comparable to that of radical anions, as commonly shown in organic molecules containing strong electron-accepting moieties, and a g-factor of 2.0027 coincides with unstable carbon radicals [35,37]. When the area of each ESR component was compared through double integration and calibration with a 2,2,6,6-tetramethylpiperidine-1-oxyl free radical (TEMPO) solution (Appendix A), **PTQ-TEG** exhibited a higher spin density than **PTQ-T**. In particular, the difference in the electrical properties of the CPs can be interpreted by comparing the g-factor 2.0027 component in each ESR signal because the g-factor of the carbon radical is comparable to that of a free electron [53]. When comparing the spin density corresponding to free electrons (g-factor of 2.0027), **PTQ-TEG** exhibited a 4.5 times higher value than **PTQ-T**. Therefore, **PTQ-TEG** has sufficient potential to exhibit higher electrical conductivity than **PTQ-T**, which could be the reason for its electrical conducting properties.

Because NO_2_ has strong electron-accepting properties, its detection generally utilizes the CT interaction with electron-donating CPs, which can generate hole carriers and increase the current level in electrical sensors, such as FET- and resistive-type electrical devices [18,37]. Because **PTQ-T** and **PTQ-TEG** exhibit semiconducting and conducting properties, respectively, their NO_2_ sensing performances were compared using an FET-type sensor for **PTQ-T** and a resistive-type sensor for **PTQ-TEG**. When the NO_2_ detection capability was evaluated by repeatedly injecting NO_2_ (50 ppm) for 50 s and N_2_ for 900 s, the source-drain current (I_DS_) of both CPs markedly increased. As shown in Figure 6a, when repeatedly exposed to NO_2_ (50 ppm) under V_G_ and V_DS_ of −10 V, **PTQ-T** in a FET-type senor exhibited a gradual increase in current change upon each NO_2_ exposure, indicating that the NO_2_ from the previous exposure did not completely escape during N_2_-filling for 900 s. The I_DS_(t)/I_DS_(0) value of **PTQ-T** did not fully recover to its initial value during N_2_ charging, indicating that NO_2_ was likely trapped in the **PTQ-T** film. In contrast, in the case of **PTQ-TEG** applied in a resistive-type sensor, the I_DS_ quickly recovered to its initial level when exposed to N_2_, demonstrating uniform detectivity during repeated NO_2_ sensing evaluations. Although NO_2_ traps in CP chains may overestimate the NO_2_ detectivity of the **PTQ-T**-based FET sensor, the NO_2_ detection capabilities of both CPs were quantified using the average values of sensitivity and recovery for three repeated NO_2_ detection experiments. **PTQ-TEG** applied to the resistive-type sensor exhibited a detectivity of 6.9%/ppm and recovery of 96%, which exceeded the detectivity and recovery of **PTQ-T** applied to the FET-type sensor (4.0%/ppm and 31%). The better NO_2_ detection characteristics of **PTQ-TEG**, compared with those of **PTQ-T**, can be attributed to the difference in the chemical structure rather than the type of electrical sensor applied to each CP. As shown in Appendix A, when the surface energies of both CPs were determined by measuring contact angles using three different solvents (water, diiodomethane, and glycerol), **PTQ-TEG** (35.16 mN/m) was confirmed to have a higher surface energy than **PTQ-T** (21.41 mN/m). This result indicates that **PTQ-TEG** has a higher affinity to polar NO_2_. In addition, compared to the alkyl side chain in **PTQ-T**, the EG-based side chain can increase the free volume of the **PTQ-TEG** film because of its more flexible nature. Indeed, **PTQ-TEG** exhibited stronger amorphous hollow diffraction than **PTQ-T** in the high-*q* region of 2D-GIXRD (Figure 3d and Appendix A). Therefore, the high affinity for NO_2_ and the enlarged free volume of **PTQ-TEG** facilitated NO_2_ diffusion into the resistive-type sensor, resulting in better NO_2_ detection ability, including faster recovery, compared to **PTQ-T**.

The selectivity and limit of detection (LOD) are important parameters of gas sensors. To demonstrate that the EG-modified polar side chain of **PTQ-TEG** did not impair the NO_2_ detection selectivity for other polar gases, the gas-sensing characteristics of the resistive sensor adopting **PTQ-TEG** were further evaluated using SO_2_, NH_3_, and CO_2_ with electron-withdrawing, electron-donating, and non-polar neutral properties, respectively. As shown in Figure 7a and Appendix A, under the same conditions as NO_2_ detection, the detection sensitivities were in the following order: NO_2_ (6.9%/ppm), CO_2_ (0.19%/ppm), NH_3_ (0.05%/ppm), and SO_2_ (0.03%/ppm). This result clearly indicates that the introduced EG-based side chains in **PTQ-TEG** only marginally impaired the selective detection of NO_2_. Interestingly, while most CP-based electrical sensors respond sensitively to NH_3_, the devised resistive sensor adopting **PTQ-TEG** exhibited a negligible response to NH_3_. Additionally, when the theoretical LOD was determined by exposure to NO_2_ at concentrations of 100, 300, 600, and 900 ppb, the **PTQ-TEG** resistive sensor responded linearly to the NO_2_ concentration (Appendix A). From the signal calibration with the signal-to-noise ratio and RMS noise, the slope extracted from the linear curve fitting using the obtained I_DS_(t)/I_DS_(0) value of each NO_2_ concentration marked a theoretical LOD of 1.59 ppb (Figure 7b). The extracted LOD was comparable to that of the most sensitive electrical NO_2_ sensor that adopted crystalline CPs with good electrical properties [31]. 

## 4. Conclusions

Two CPs with the same conjugated backbone containing different side chains were designed to investigate the effect of the side chain on their electrical properties. Although both CPs exhibited similar smooth morphologies and weak crystalline chain assemblies in the film state, they exhibited completely different electrical properties. Specifically, **PTQ-T** with alkyl side chains showed typical *p*-type semiconducting characteristics, whereas **PTQ-TEG** with EG-based side chains exhibited electrical conducting behaviors. It was confirmed that, although both CPs have radical species owing to their strong D-A type conjugated structure, the O atoms of the EG-based side chains can additionally intercalate with the conjugated backbone, increase the carrier density, and ultimately generate the conductor-like properties of **PTQ-TEG**. When **PTQ-T** and **PTQ-TEG** were applied to FET- and resistive-type sensors, respectively, **PTQ-TEG** exhibited higher NO_2_ sensitivity with a faster recovery tendency than **PTQ-T**. The flexible EG-based side chains increased the free volume of the CP chains as well as the affinity with polar NO_2_ molecules, which facilitated NO_2_ diffusion in and out of the **PTQ-TEG** film, resulting in better sensitivity to NO_2_ than **PTQ-T**. In addition, it was confirmed that the EG-based side chains in **PTQ-TEG** barely impaired the detection selectivity for other common gases, such as SO_2_, NH_3_, and CO_2_.

## Figures and Tables

**Figure 1 materials-16-02877-f001:**
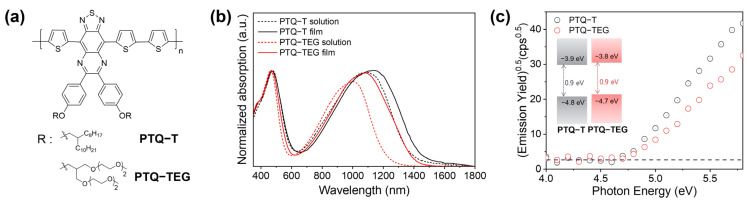
(**a**) Chemical structures; (**b**) absorption tendencies, and (**c**) electronic energy levels of the obtained CPs.

**Figure 2 materials-16-02877-f002:**
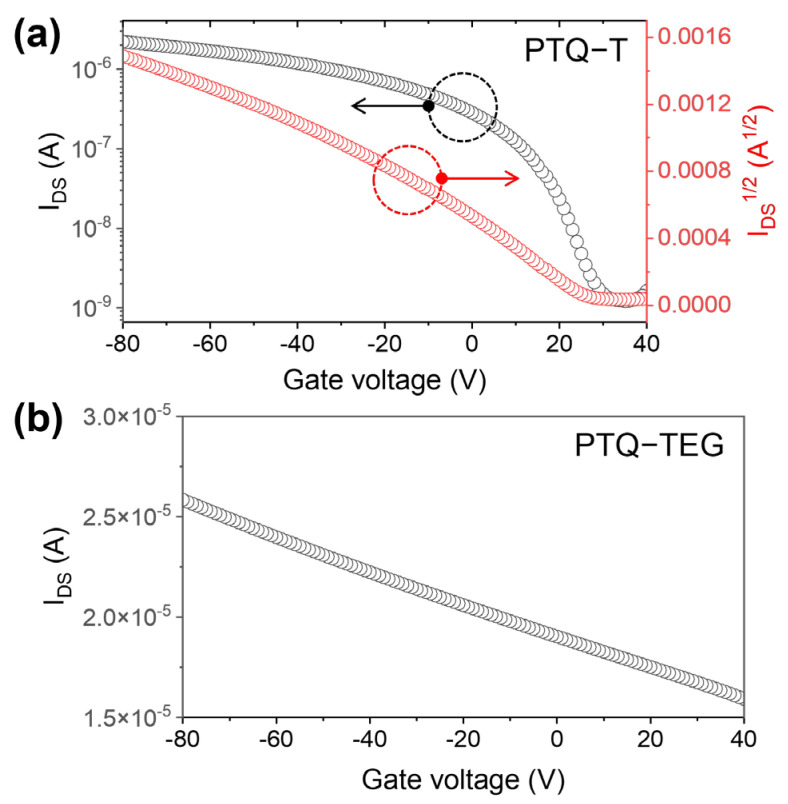
FET curves of (**a**) **PTQ-T** and (**b**) **PTQ-TEG**.

**Figure 3 materials-16-02877-f003:**
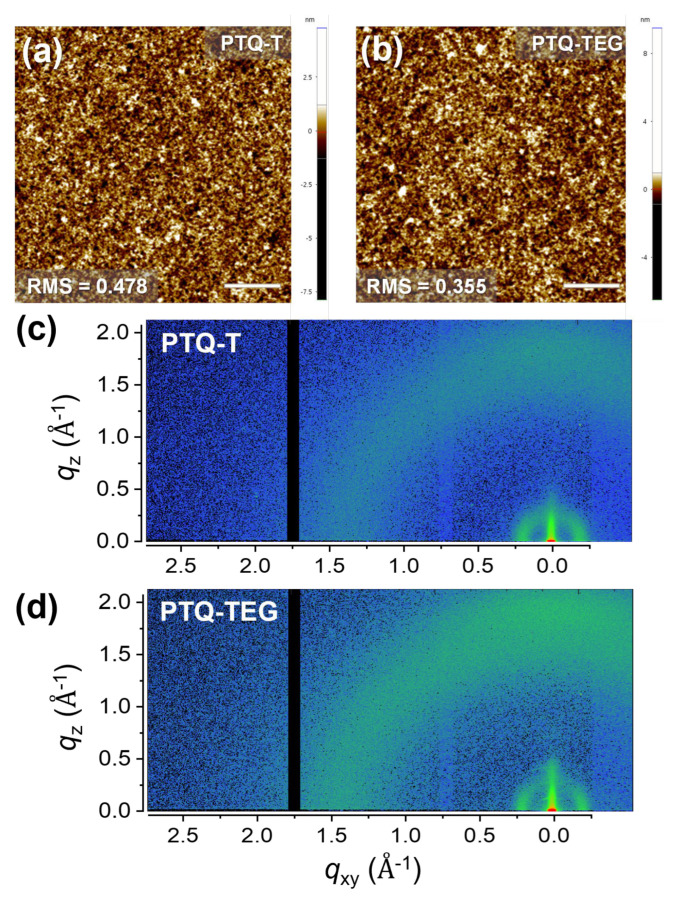
Surface morphologies and chain assembly tendencies characterized using AFM and 2D-GIXRD for (**a**,**c**) **PTQ-T** and (**b**,**d**) **PTQ-TEG**.

**Figure 4 materials-16-02877-f004:**
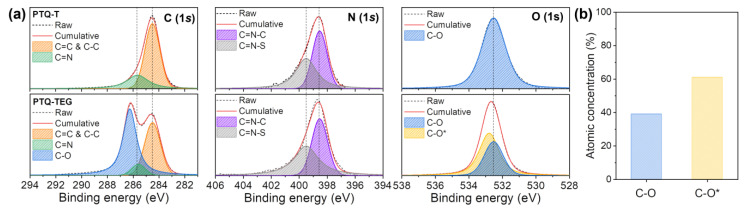
(**a**) Atomic binding-energy distributions of **PTQ-T** and **PTQ-TEG** and (**b**) comparison of intercalated O atoms in **PTQ-TEG**. “*” indicates the presence of O atoms having a higher binding energy compared to the pristine O atoms.

**Figure 5 materials-16-02877-f005:**
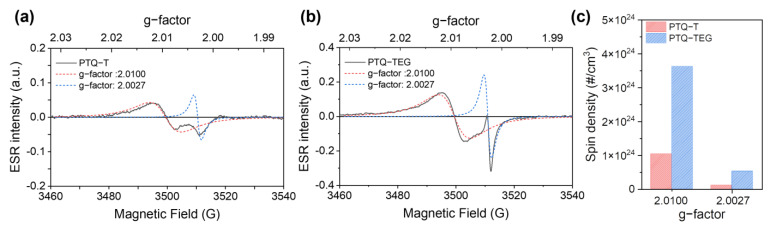
ESR signals and corresponding g-factors of (**a**) **PTQ-T** and (**b**) **PTQ-TEG** and comparison of the spin density of each ESR component. (**c**) indicates the spin density corresponding to each g-factor in ESR signals of both CPs.

**Figure 6 materials-16-02877-f006:**
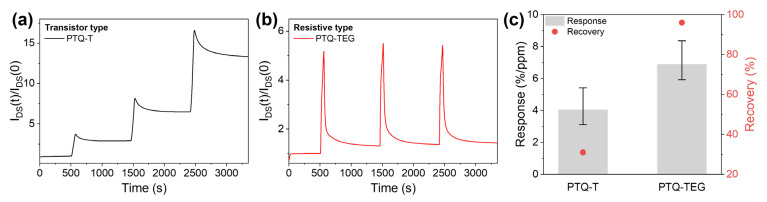
NO_2_ detection characteristics of (**a**) **PTQ-T** and (**b**) **PTQ-TEG** and comparison of their sensitivity and recovery tendency. (**c**) indicates the NO_2_ detectivity and recovery tendency of each CP.

**Figure 7 materials-16-02877-f007:**
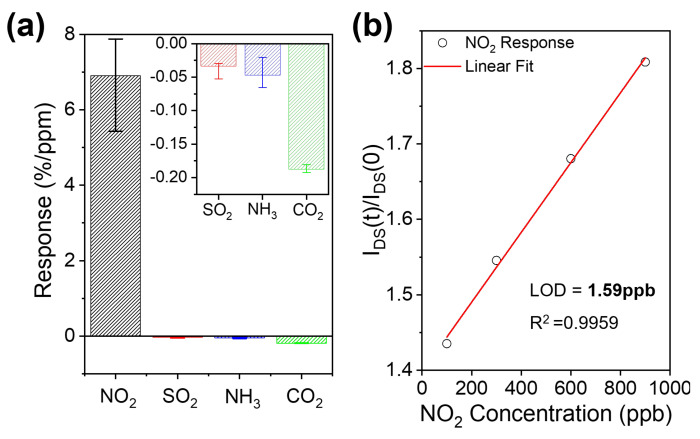
(**a**) Detection selectivity and (**b**) the calculated theoretical LOD of a resistive sensor adopting **PTQ-TEG**.

## Data Availability

Data sharing not applicable, all data obtained from this study are already given in the article.

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
