# Peer review of "Side-Chain-Assisted Transition of Conjugated Polymers from a Semiconductor to Conductor and Comparison of Their NO2 Sensing Characteristics"

_materials, 2023, doi:10.3390/ma16072877_

Round 1
Reviewer 1 Report
In this manuscript, B. G. Kim et al. reported the synthesis of two donor-acceptor conjugated polymers containing aliphatic and ethylene oxide derivative side chains on their conjugated backbone (PTQ-T and PTQ-TEG). For application, PTQ-TEG exhibits a better NO2 detection capability with faster signal recovery characteristics than PTQ-T. The subject is interesting and the paper could be considered as good. Some minor revisions are needed before final acceptance.
1- The spectroscopy analyses to confirm the chemical structure of two CPs are missing. FTIR and H-NMR spectroscopy for PTQ-T and PTQ-TEG polymers need to provide.
2- The conditions of the UV-vis adsorption experiments are missing. Which solvent was used to measure samples in solution states. In addition, “although the absorption peaks of PTQ-T containing aliphatic side chains tended to be more red-shifted, both CPs exhibited typical bimodal-shaped absorption spectra”, this explanation is not clear. The authors need to re-explain with references.
3- The authors mentioned that “Interestingly, the absorption in the film state was found to be more red-shifted than that in the solution state only in the longer wavelength
region. This can be attributed to the facile intramolecular CT interaction between D-A
moieties through partial chain planarization of the CPs in the film state”. How about the non-covalent interactions between the side-chains and solvents? Usually side-chins have strong effect of the properties of polymers (https://doi.org/10.1021/acsapm.2c01460).
4- In XPS, there are two kinds of nitrogens; C=N-C and C=N-S and the fitting of the sulfur atom is missing. Please, revise the XPS.
5- In the first paragraph of the introduction section, there are many previous works published for the preparation of conjugated donor-acceptor conjugated microporous polymers and their advanced properties. The authors need to acknowledge the previous literature and compare their work with the conjugated microporous donor-acceptor polymers in the literature and demonstrate their research outcomes in terms of advantages and disadvantages. Some of studies are given below need to cited at suitable places; https://doi.org/10.1002/cctc.202201287; https://doi.org/10.1021/jacs.1c07916; https://doi.org/10.1016/j.apcatb.2022.121624; https://doi.org/10.1016/j.micromeso.2022.112258; https://doi.org/10.1016/j.cej.2023.141553.
6- The authors need to study the regeneration and chemical stability of the PTQ-TEG after NO2 detection.
Reviewer 2 Report
In this paper, the author synthesized two polymers with different side chains and exhibited electrical properties. PTQ-TEG exhibited higher NO2 sensitivity with a faster recovery tendency than PTQ-T. The authors need to review the paper and realize more about the research on ethylene glycol side chains, provide reasonable explanations for two polymers with the different side chain. This paper can be accepted after corrections.
1. In line 10, it is suggested to change “aliphatic” and “ethylene oxide” to “alkyl” and “ethylene glycol” in this paper.
2. In line 17 and 23, is it the field-effect transistor or field-effect sensor?
3. The grammar should be revised through the whole paper, such as line 37 to 39, 45 to 46......
4. It is suggested to describe the advantages and related research of ethylene glycol side chains and indicates that the introduction of ethylene glycol has a significant effect for the polymer. For instance, conjugated polymers with branched ethylene glycol side chains for organic thermoelectrics. (https://onlinelibrary.wiley.com/doi/10.1002/anie.202214192) .
5. Line 61, change “onto” to “into”.
6. Illustrate the polymers corresponding to Figure S1 and S2, check supporting information carefully.
7. UV-vis absorption, explain the red-shifted absorption of polymers with different side chains.
8. Line 174 to 175, “typical p-type charge-transport behavior” and “the electrical mobility” are contradictory.
9. Line 181 and 183, revise “S/cm”.
Reviewer 3 Report
This article, “Side-Chain-Assisted Transition of Conjugated Polymers from a Semiconductor to Conductor and Comparison of their NO2 Sensing Characteristics," explores how the addition of side chains in conjugated polymers (CPs) affects their electrical properties. Two CPs were created with aliphatic and ethylene oxide (EO) derivative side chains on the same conjugated backbone. PTQ-T with an aliphatic side chain was a p-type semiconductor while PTQ-TEG with an EO-based side chain exhibited electrical conductivity and a higher potential for NO2 detection. X-ray photoelectron spectroscopy analysis showed that the O atoms of the EO-based side chains in PTQ-TEG were intercalated with the conjugated backbone, increasing carrier density. PTQ-TEG showed better NO2 detection capability with faster signal recovery than PTQ-T in resistive sensors and field-effect transistors due to the flexible EO-based side-chains improving NO2-affinity and enlarging free volume. The findings are helpful to the polymer community. Therefore, I would recommend it be published in Materials after the following minor issues are addressed:
- The authors are encouraged to provide the thickness of the measurement samples, including AFM, device, and 2D-GIXRD, to increase the reproducibility and transparency of the experimental results.
- The authors should provide the yields of the polymers to improve the clarity and reliability of the synthetic procedures.
- The authors are advised to provide a more detailed explanation of the UV-Vis difference between the two polymers, given that the absorption should be more similar in regular side-chain engineering of polymers. As the feature of this work is the proper selection of EG side chains, the authors should consider providing enough explanation for clarity.
- For the analysis of 2D-GIXRD, it would be helpful if the authors provide the 1-D line cut figures to quantitatively compare the peaks for morphological study. The authors concluded that flexible EO-based side chains increased the free volume, making 2D-GIXRD an important clue.
- On page 6, line 204, it is suggested that the authors correct "small-angle region" to "high-q region."
- The authors should consider measuring the surface energies of the polymers to support the affinity perspective.
- The introduction could be improved by reorganizing the content to strengthen the motivation of the work. For example, sentences mentioned in the discussion (such as on page 7, line 273-275 and page 8, line 298-300) should be introduced in the introduction.
- For the introduction, several works of alkoxy side chain engineered polymer under similar backbone structures should be cited — [J. Mater. Chem. A, 2019,7, 3072-3082; Organic Electronics 2020, 87, 105986. https://doi.org/10.1016/j.orgel.2020.105986.; Molecules 2023, 28(5), 2056; https://doi.org/10.3390/molecules28052056; Molecules 2022, 27(23), 8424 (doi.org/10.3390/molecules27238424)
